# $q$-*OCSVM*: A $q$-**Quantile Estimator for High-Dimensional Distributions**

**Assaf Glazer**      **Michael Lindenbaum**      **Shaul Markovitch**
Department of Computer Science, Technion - Israel Institute of Technology
{assafgr,mic,shaulm}@cs.technion.ac.il

## Abstract

In this paper we introduce a novel method that can efficiently estimate a family of hierarchical dense sets in high-dimensional distributions. Our method can be regarded as a natural extension of the *one-class SVM (OCSVM)* algorithm that finds multiple parallel separating hyperplanes in a reproducing kernel Hilbert space. We call our method $q$-*OCSVM*, as it can be used to estimate $q$ quantiles of a high-dimensional distribution. For this purpose, we introduce a new global convex optimization program that finds all estimated sets at once and show that it can be solved efficiently. We prove the correctness of our method and present empirical results that demonstrate its superiority over existing methods.

## 1   Introduction

*One-class SVM (OCSVM)* [14] is a kernel-based learning algorithm that is often considered to be the method of choice for set estimation in high-dimensional data due to its generalization power, efficiency, and nonparametric nature. Let $\mathcal{X}$ be a training set of examples sampled i.i.d. from a continuous distribution $F$ with Lebesgue density $f$ in $\mathbb{R}^d$. The *OCSVM* algorithm takes $\mathcal{X}$ and a parameter $0 < \nu < 1$, and returns a subset of the input space with a small volume while bounding a $\nu$ portion of examples in $\mathcal{X}$ outside the subset. Asymptotically, the probability mass of the returned subset converges to $\alpha = 1 - \nu$. Furthermore, when a Gaussian kernel with a zero tending bandwidth is used, the solution also converges to the *minimum-volume set (MV-set)* at level $\alpha$ [19], which is a subset of the input space with the smallest volume and probability mass of at least $\alpha$.

In light of the above properties, the popularity of the *OCSVM* algorithm is not surprising. It appears, however, that in some applications we are not actually interested in estimating a single MV-set but in estimating multiple hierarchical MV-sets, which reveal more information about the distribution. For instance, in cluster analysis [5], we are interested in learning hierarchical MV-sets to construct a cluster tree of the distribution. In outlier detection [6], hierarchical MV-sets can be used to classify examples as outliers at different levels of significance. In statistical tests, hierarchical MV-sets are used for generalizing univariate tests to high-dimensional data [12, 4]. We are thus interested in a method that generalizes the *OCSVM* algorithm for approximating hierarchical MV-sets. By doing so we would leverage the advantages of the *OCSVM* algorithm in high-dimensional data and take it a step forward by extending its solution for a broader range of applications.

Unfortunately, a straightforward approach of training a set of *OCSVMs*, one for each MV-set, would not necessarily satisfy the hierarchy requirement. Let $q$ be the number of hierarchical MV-sets we would like to approximate. A naive approach would be to train $q$ *OCSVMs* independently and enforce hierarchy by intersection operations on the resulting sets. However, we find two major drawbacks in this approach: (1) The $\nu$-property of the *OCSVM* algorithm, which provides us with bounds on the number of examples in $\mathcal{X}$ lying outside or on the boundary of each set, is no longer guaranteed due to the intersection operator; (2) MV-sets of a distribution, which are also level sets of the distribution's density $f$ (under sufficient regularity conditions), are hierarchical by definition. Hence,

by learning $q$ *OCSMs* independently, we ignore an important property of the correct solution, and thus are less likely reach a generalized global solution.

In this paper we introduce a generalized version of the *OCSVM* algorithm for approximating hierarchical MV-sets in high-dimensional distributions. As in the naive approach, approximated MV-sets in our method are represented as dense sets captured by separating hyperplanes in a reproducing kernel Hilbert space. However, our method does not suffer from the two drawbacks mentioned above. To preserve the $\nu$-property of the solution while fulfilling the hierarchy constraint, we require the resulting hyperplanes to be parallel to one another. To provide a generalized global solution, we introduce a new convex optimization program that finds all approximated MV-sets at once. Furthermore, we expect our method to have better generalization ability due to the parallelism constraint imposed on the hyperplanes, which also acts as a regularization term on the solution.

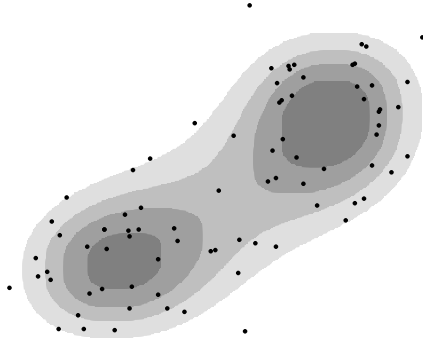

***Figure 1:*** An approximation of 4 hierarchical MV-sets

We call our method $q$-*OCSVM*, as it can be used by statisticians to generalize $q$-quantiles to high-dimensional distributions. Figure 1 shows an example of $4$-quantiles estimated for two-dimensional data. We show that our method can be solved efficiently, and provide theoretical results showing that it preserves both the density assumption for each approximated set in the same sense suggested by [14]. In addition, we empirically compare our method to existing methods on a variety of real high-dimensional data and show its advantages in the examined domains.

## 2 Background

In one-dimensional settings, $q$-quantiles, which are points dividing a cumulative distribution function (*CDF*) into equal-sized subsets, are widely used to understand the distribution of values. These points are well defined as the inverse of the *CDF*, that is, the quantile function. It would be useful to have the same representation of $q$-quantiles in high-dimensional settings. However, it appears that generalizing quantile functions beyond one dimension is hard since the number of ways to define them grows exponentially with the dimensions [3]. Furthermore, while various quantile regression methods [7, 16, 9] can be to used to estimate a single quantile of a high-dimensional distribution, extensions of those to estimate $q$-quantiles is mostly non-trivial.

Let us first understand the exponential complexity involved in estimating a generalized quantile function in high-dimensional data. Let $0 < \alpha_1 < \alpha_2, \ldots, < \alpha_q < 1$ be a sequence of equally-spaced $q$ quantiles. When $d = 1$, the quantile transforms $F^{-1}(\alpha_j)$ are uniquely defined as the points $x_j \in \mathbb{R}$ satisfying $F(X \leq x_j) \leq \alpha_j$, where $X$ is a random variable drawn from $F$. Equivalently, $F^{-1}(\alpha_j)$ can be identified with the unique hierarchical intervals $[-\infty, x_j]$. However, when $d > 1$, intervals are replaced by sets $C_1 \subset C_2 \ldots \subset, C_q$ that satisfy $F(C_j) = \alpha_j$ but are not uniquely defined. Assume for a moment that these sets are defined only by imposing directions on $d - 1$ dimensions (the direction of the first dimension can be chosen arbitrarily). Hence, we are left with $2^{d-1}$ possible ways of defining a generalized quantile function for the data.

Hypothetically, any arbitrary hierarchical sets satisfying $F(C_j) = \alpha_j$ can be used to define a valid generalized quantile function. Nevertheless, we would like the distribution to be dense in these sets so that the estimation will be informative enough. Motivated in this direction, Polonik [12] suggested using hierarchical MV-sets to generalize quantile functions. Let $C(\alpha)$ be the MV-set at level $\alpha$ with respect to $F$ and the Lebesgue density $f$. Let $L_f(c) = \{x \ : \ f(x) \geq c\}$ be the *level set*

of $f$ at level $c$. Polonik observed that, under sufficient regularity conditions on $f$, $L_f(c)$ is an MV-set of $F$ at level $\alpha = F(L_f(c))$. He thus suggested that level sets can be used as approximations of the MV-sets of a distribution. Since level sets are hierarchical by nature, a density estimator over $\mathcal{X}$ would be sufficient to construct a generalized quantile function.

Polonik's work was largely theoretical. In high-dimensional data, not only is the density estimation hard, but extracting level sets of the estimated density is also not always feasible. Furthermore, in high-dimensional settings, even attempting to estimate $q$ hierarchical MV-sets of a distribution might be too optimistic an objective due to the exponential growth in the search space, which may lead to overfitted estimates, especially when the sample is relatively small. Consequently, various methods were proposed for estimating $q$-quantiles in multivariate settings without an intermediate density estimation step [3, 21, 2, 20]. However, these methods were usually efficient only up to a few dimensions. For a detailed discussion about generalized quantile functions, see Serfling [15].

One prominent method that uses a variant of the *OCSVM* algorithm for approximating level sets of a distribution was proposed by Lee and Scott [8]. Their method is called *nested OCSVM (NOC-SVM)* and it is based on a new quadratic program that simultaneously finds a global solution of multiple nested half-space decision functions. An efficient decomposition method is introduced to solve this program for large-scale problems. This program uses the *C*-SVM formulation of the *OCSVM* algorithm [18], where $\nu$ is replaced by a different parameter, $C \geq 0$, and incorporates nesting constraints into the dual quadratic program of each approximated function. However, due to these difference formulations, our method converges to predefined $q$-quantiles of a distribution while theirs converges to approximated sets with unpredicted probability masses. The probability masses in their solution are even less trackable because the constraints imposed by the *NOC-SVM* program on the dual variables changes the geometric interpretation of the primal variables in a non-intuitive way. An improved quantile regression variant of the *OCSVM* algorithm that also uses "non-crossing" constraints to estimate "non-crossing" quantiles of a distribution was proposed by Takeuchi et al. [17]. However, similar to the *NOC-SVM* method, after enforcing these constraints, the $\nu$-property of the solution is no longer guaranteed.

Recently, a greedy *hierarchical MV-set estimator (HMVE)* that uses *OCSVMs* as a basic component was introduced by Glazer et al. [4]. This method approximates the MV-sets iteratively by training a sequence of *OCSVMs*, from the largest to the smallest sets. The superiority of *HMVE* was shown over a density-based estimation method and over a different hierarchical MV-set estimator that was also introduced in that paper and is based on the one-class neighbor machine (*OCNM*) algorithm [11]. However, as we shall see in experiments, it appears that approximations in this greedy approach tend to become less accurate as the required number of MV-sets increases, especially for approximated MV-sets with small $\alpha$ in the last iterations.

In contrast to the naive approach of training $q$ *OCSVMs* independently [1], our $q$-*OCSVM* estimator preserves the $\nu$-property of the solution and converges to a generalized global solution. In contrast to the *NOC-SVM* algorithm, $q$-*OCSVM* converges to predefined $q$-quantiles of a distribution. In contrast to the *HMVE* estimator, $q$-*OCSVM* provides global and stable solutions. As will be seen, we support these advantages of our method in theoretical and empirical analysis.

## 3   The $q$-*OCSVM* Estimator

In the following we introduce our $q$-*OCSVM* method, which generalizes the *OCSVM* algorithm so that its advantages can be applied to a broader range of applications. $q$ stands for the number of MV-sets we would like our method to approximate.

Let $\mathcal{X} = \{x_1, \ldots, x_n\}$ be a set of feature vectors sampled i.i.d. with respect to $F$. Consider a function $\Phi : \mathbb{R}^d \to \mathcal{F}$ mapping the feature vectors in $\mathcal{X}$ to a hypersphere in an infinite Hilbert space $\mathcal{F}$. Let $\mathcal{H}$ be a hypothesis space of half-space decision functions $f_C(x) = sgn\left((w \cdot \Phi(x)) - \rho\right)$ such that $f_C(x) = +1$ if $x \in C$, and $-1$ otherwise. The *OCSVM* algorithm returns a function $f_C \in \mathcal{H}$ that maximizes the margin between the half-space decision boundary and the origin in $\mathcal{F}$, while bounding a portion of examples in $\mathcal{X}$ satisfying $f_C(x) = -1$. This bound is predefined by a parameter $0 < \nu < 1$, and it is also called the $\nu$-property of the *OCSVM* algorithm. This function is

specified by the solution of this quadratic program:

$$\min_{w\in\mathcal{F},\xi\in\mathbb{R}^n,\rho\in\mathbb{R}}\frac{1}{2}||w||^2 - \rho + \frac{1}{\nu n}\sum_i \xi_i, \quad s.t. \quad (w\cdot\Phi(x_i))\geq\rho-\xi_i, \; \xi_i\geq 0, \quad (1)$$

where $\xi$ is a vector of the slack variables. All training examples $x_i$ for which $(w\cdot\Phi(x))-\rho\leq 0$ are called *support vectors (SVs)*. Outliers are referred to as examples that strictly satisfy $(w\cdot\Phi(x))-\rho < 0$. By solving the program for $\nu = 1-\alpha$, we can use the *OCSVM* to approximate $C(\alpha)$.

Let $0 < \alpha_1 < \alpha_2, \ldots, < \alpha_q < 1$ be a sequence of $q$ quantiles. Our goal is to generalize the *OCSVM* algorithm for approximating a set of MV-sets $\{C_1, \ldots, C_q\}$ such that a hierarchy constraint $C_i \subseteq C_j$ is satisfied for $i < j$. Given $\mathcal{X}$, our *q-OCSVM* algorithm solves this primal program:

$$\min_{w,\xi_j,\rho_j}\frac{q}{2}||w||^2 - \sum_{j=1}^{q}\rho_j + \sum_{j=1}^{q}\frac{1}{\nu_j n}\sum_i \xi_{j,i}$$

$$s.t. \quad (w\cdot\Phi(x_i))\geq\rho_j-\xi_{j,i}, \; \xi_{j,i}\geq 0, \; j\in[q], i\in[n], \quad (2)$$

where $\nu_j = 1-\alpha_j$. This program generalizes Equation (1) to the case of finding multiple, parallel half-space decision functions by searching for a global minimum over their sum of objective functions: the coupling between $q$ half-spaces is done by summing $q$ *OCSVM* programs, while enforcing these programs to share the same $w$. As a result, the $q$ half-spaces in the solution of Equation (2) are different only by their bias terms, and thus parallel to each other. This program is convex, and thus a global minimum can be found in polynomial time.

It is important to note that even with an ideal, unbounded number of examples, this program does not necessarily converge to the exact MV-sets but to approximated MV-sets of a distribution. As we shall see in Section 4, all decision functions returned by this program preserve the $\nu$-property. We argue that the stability of these approximated MV-sets benefits from the parallelism constraint imposed on the half-spaces in $\mathcal{H}$, which acts as a regularizer.

In the following we show that our program can be solved efficiently in its dual form. Using multipliers $\eta_{j,i}\geq 0, \beta_{j,i}\geq 0$, the Lagrangian of this program is

$$L(\boldsymbol{w},\boldsymbol{\xi_q},\boldsymbol{\rho_1},\ldots,\boldsymbol{\rho_q},\eta,\beta) = \frac{q}{2}||w||^2 - \sum_{j=1}^{q}\rho_j + \sum_{j=1}^{q}\frac{1}{\nu_j n}\sum_i \xi_{j,i}$$

$$-\sum_{j=1}^{q}\sum_i \eta_{j,i}((\Phi(x_i)\cdot w_j)-\rho_j+\xi_{j,i}) - \sum_{j=1}^{q}\sum_i \beta_{j,i}\xi_{j,i}. \quad (3)$$

Setting the derivatives to be equal to zero with respect to the primal variables $w,\rho_j,\xi_j$ yields

$$w = \frac{1}{q}\sum_{j,i}\eta_{j,i}\Phi(x_i), \quad \sum_i\eta_{j,i}=1, \; 0\leq\eta_{ji}\leq\frac{1}{n\nu_j}, \; i\in[n], j\in[q]. \quad (4)$$

Substituting Equation (4) into Equation (3), and replacing the dot product $(\Phi(x_i)\cdot\Phi(x_s))_{\mathcal{F}}$ with a kernel function $k(x_i,x_s)$ [2], we obtain the dual program

$$\min_{\eta}\frac{1}{2q}\sum_{j,p\in[q]}\sum_{i,s\in[n]}\eta_{j,i}\eta_{p,s}k(x_i,x_s), \quad s.t.\sum_i\eta_{j,i}=1, \; 0\leq\eta_{ji}\leq\frac{1}{n\nu_j}, \; i\in[n], j\in[q]. \quad (5)$$

Similar to the formulation of the dual objective function in the original *OCSVM* algorithm, our dual program depends only on the $\eta$ multipliers, and hence can be solved more efficiently than the primal one. The resulting decision function for $j$'th estimate is

$$f_{C_j}(x) = \text{sgn}\left(\frac{1}{q}\sum_i\eta_i^* k(x_i,x) - \rho_j\right), \quad (6)$$

where $\eta_i^* = \sum_{j=1}^q \eta_{j,i}$. This efficient formulation of the decision function, which derives from the fact that parallel half-spaces share the same $w$, allows us to compute the outputs of all the $q$ decision functions simultaneously.

As in the *OCSVM* algorithm, $\rho_j$ are recovered by identifying points $\Phi(x_{j,i})$ lying strictly on the $j$'th decision boundary. These points are identified using the condition $0 < \eta_{j,i} < \frac{1}{n\nu_j}$. Therefore, $\rho_j$ can be recovered from a point $sv$ satisfying this condition by

$$\rho_j = (w \cdot \Phi(sv)) = \frac{1}{q} \sum_i \eta_i^* k(x_i, sv). \tag{7}$$

Figure 1 shows the resulting estimates of our $q$-*OCSVM* method for 4 hierarchical MV-sets with $\alpha = 0.2, 0.4, 0.6, 0.8$ [3]. 100 train examples drawn i.i.d. from a bimodal distribution are marked with black dots. It can be seen that the number of bounded SVs (outliers) at each level is no higher than $100(1 - \alpha_j)$, as expected according to the properties of our $q$-*OCSVM* estimator, which will be proven in the following section.

## 4    Properties of the $q$-*OCSVM* Estimator

In this section we provide theoretical results for the $q$-*OCSVM* estimator. The program we solve is different from the one in Equation (1). Hence, we can not rely on the properties of *OCSVM* to prove the properties of our method. We provide instead similar proofs, in the spirit of Schölkopf et al. [14] and Glazer et al. [4], with some additional required extensions.

**Definition 1.** *A set* $\mathcal{X} = \{x_1, \ldots, x_n\}$ *is* separable *if there exists some $w$ such that $(\Phi(x_i) \cdot w) > 0$ for all $i \in \{1, \ldots, n\}$.*

Note that if a Gaussian kernel is used (implies $k(x_i, x_s) > 0$), as in our case, then $\mathcal{X}$ is separable.

**Theorem 1.** *If $\mathcal{X}$ is separable, then a feasible solution exists for Equation (2) with $\rho_j > 0$ for all $j \in \{1, \ldots, q\}$.*

*Proof.* Define $M$ as the convex hull of $\Phi(x_1), \cdots, \Phi(x_n)$. Note that since $\mathcal{X}$ is separable, $M$ does not contain the origin. Then, by the supporting hyperplane theorem [10], there exists a hyperplane $(\Phi(x_i) \cdot w) - \rho$ that contains $M$ on one side of it and does not contain the origin. Hence, $\left(\overrightarrow{0} \cdot w\right) - \rho < 0$, which leads to $\rho > 0$. Note that the solution $\rho_j = \rho$ for all $j \in [q]$ is a feasible solution for Equation (2). $\square$

The following theorem shows that the regions specified by the decision functions $f_{C_1}, \ldots, f_{C_q}$ are (a) approximations for the MV-sets in the same sense suggested by Schölkopf et al., and (b) hierarchically nested.

**Theorem 2.** *Let $f_{C_1}, \ldots, f_{C_q}$ be the decision functions returned by the $q$-OCSVM estimator with parameters $\{\alpha_1, \ldots, \alpha_q\}, \mathcal{X}, k(\cdot, \cdot)$. Assume $\mathcal{X}$ is separable. Let $SV_{o_j}$ be the set of SVs lying strictly outside $C_j$, and $SV_{b_j}$ be the set of SVs lying exactly on the boundary of $C_j$. Then, the following statements hold:(1) $C_j \subseteq C_k$ for $\alpha_j < \alpha_k$. (2) $\frac{|SV_{o_j}|}{|\mathcal{X}|} \le 1 - \alpha_j \le \frac{|SV_{b_j}| + |SV_{o_j}|}{|\mathcal{X}|}$. (3) Suppose $\mathcal{X}$ is i.i.d. drawn from a distribution $F$ which does not contain discrete components, and $k(\cdot, \cdot)$ is analytic and non-constant. Then, $\frac{|SV_{o_j}|}{|\mathcal{X}|}$ is asymptotically equal to $1 - \alpha_j$.*

*Proof.* $C_j$ and $C_k$ are associated with two parallel half-spaces in $\mathcal{H}$ with the same $w$. Therefore, statement (1) can be proven by showing that $\rho_j \ge \rho_k$. $\alpha_j < \alpha_k$ leads to $\rho_j \ge \rho_k$ since otherwise the optimality of Equation (2) would be contradicted. Assume by *negation* that $\nu_j = 1 - \alpha_j > \frac{|SV_{b_j}| + |SV_{o_j}|}{|\mathcal{X}|}$ for some $j \in [q]$ in the optimal solution of Equation (2). Note that when parallel-shifting the optimal hyperplane by slightly increasing $\rho_j$, the term $\sum_i \xi_{j,i}$ in the equation will change proportionally to $|SV_{b_j}| + |SV_{o_j}|$. However, since $\frac{|SV_{b_j}| + |SV_{o_j}|}{|\mathcal{X}|\nu_j} < 1$, a slight increase in $\rho_j$ will

result in a decrease in the objective function, which contradicts the optimality of the hyperplane. The same goes for the other direction: Assume by *negation* that $\frac{|SV_{o_j}|}{|\mathcal{X}|} > 1 - \alpha_j$ for some $j \in [q]$ in the optimal solution of Equation (2). Then, a slight decrease in $\rho_j$ will result in a decrease in the objective function, which contradicts the optimality of the hyperplane. We are now left to prove statement (3): The covering number of the class of $f_{C_j}$ functions (which are induced by $k$) is well-behaved. Hence, asymptotically, the probability of points lying exactly on the hyperplanes converges to zero (cf. 13). $\qquad\qquad\square$

## 5 Empirical Results

We extensively evaluated the effectiveness of our *q-OCSVM* method on a variety of real high-dimensional data from the UCI repository and the 20-Newsgroup document corpus, and compared its performance to competing methods.

### 5.1 Experiments on the UCI Repository

We first evaluated our method on datasets taken from the UCI repository [4]. From each examined dataset, a random set of 100 examples from the most frequent label was used as the training set $\mathcal{X}$. The remaining examples from the same label were used as the test set. We used all UCI datasets with more than 50 test examples — a total of 61 data sets. The average number of features for a dataset is 113 [5].

We compared the performance of our *q-OCSVM* method to three alternative methods that generalize the *OCSVM* algorithm: *HMVE (hierarchical minimum-volume estimator)* [4], *I-OCSVM (independent one-class SVMs)*, and *NOC-SVM (nested one-class SVM)* [8]. For the *NOC-SVM* method, we used the implementation provided by the authors [6]. The *LibSVM* package [1] was used to implement the *HMVE* and *I-OCSVM* methods. An implementation of our *q-OCSVM* estimator is available from: `http://www.cs.technion.ac.il/˜assafgr/articles/q-ocsvm.html`. All experiments were carried out with a Gaussian kernel ($\gamma = \frac{1}{2\sigma^2} = \frac{2.5}{\#features}$).

For each data set, we trained the reference methods to approximate hierarchical MV-sets at levels $\alpha_1 = 0.05, \alpha_2 = 0.1 \ldots, \alpha_{19} = 0.95$ (19-quantiles) [7]. Then, we evaluated the estimated $q$-quantiles with the test set. Since the correct MV-sets are not known for the data, the quality of the approximated MV-sets was evaluated by the *coverage ratio (CR)*: Let $\alpha'$ be the empirical proportion of the approximated MV-sets that was measured with the test data. The expected proportion of examples that lie within the MV-set $C(\alpha)$ is $\alpha$. The coverage ratio is defined as $\frac{\alpha'}{\alpha}$. A perfect MV-set approximation method would yield a coverage ratio of 1.0 for all approximated MV-sets [8]. An advantage of choosing this measure for evaluation is that it gives more weight for differences between $\alpha$ and $\alpha'$ in small quantiles associated with regions of high probability mass.

Results on test data for each approximated MV-set are shown in Figure 2. The left graph displays in bars the empirical proportion of test examples in the approximated MV-sets ($\alpha'$) as a function of the expected proportion ($\alpha$) averaged over all 61 data sets. The right graph displays the coverage ratio of test examples as a function of $\alpha$ averaged over all 61 data sets. It can be seen that our *q-OCSVM* method dominates the others with the best average $\alpha'$ and average coverage ratio behaviors. In each quantile separately, we tested the significance of the advantage of *q-OCSVM* over the competitors using the Wilcoxon statistical test over the absolute difference between the expected and empirical coverage ratios ($|1.0 - CR|$). The superiority of our method against the three competitors was found significant, with $P < 0.01$, for each of the 19 quantiles separately.

The *I-OCSVM* method shows inferior performance to that of *q-OCSVM*. We ascribe this behavior to the fact that it trains $q$ *OCSVMs* independently, and thus reaches a local solution. Furthermore, we

believe that by ignoring the fundamental hierarchical structure of MV-sets, the *I-OCSVM* method is more likely than ours to reach an overfitted solution.

The *HMVE* method shows a decrease in performance from the largest to the smallest $\alpha$. We assume this is due to the greedy nature of this method. *HMVE* approximates the MV-sets iteratively by training a sequence of *OCSVMs*, from the largest to the smallest $\alpha$ . *OCSVMs* trained later in the sequence are thus more constrained in their approximations by solutions from previous iterations, so that the error in approximations accumulates over time. This is in contrast to *q-OCSVM*, which converges to a global minimum, and hence is more scalable than *HMVE* with respect to the number of approximated MV-sets ($q$). The *NOC-SVM* method performs poorly in comparison to the other methods. This is not surprising, since, unlike the other methods, we cannot set the parameters of *NOC-SVM* to converge to predefined $q$-quantiles.

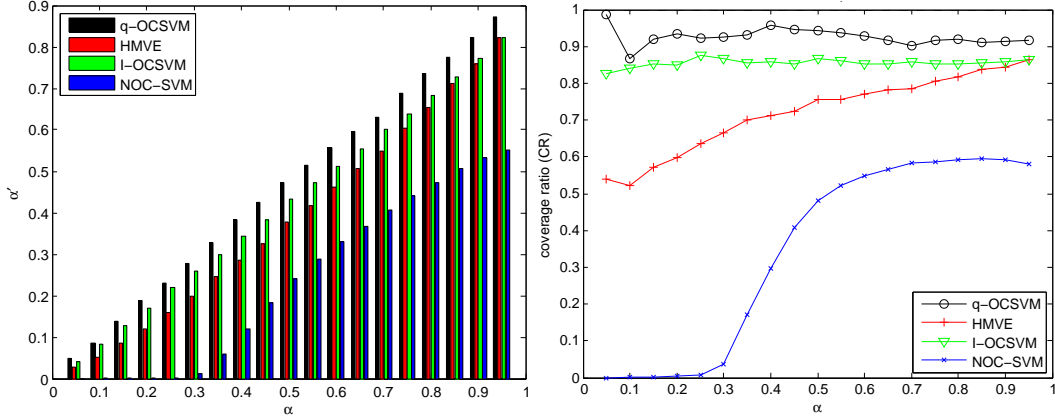

***Figure 2:*** The *q-OCSVM*, *HMVE*, *I-OCSVM*, and *NOC-SVM* methods were trained to estimate 19-quantiles for the distribution of the most frequent label on the 61 UCI datasets. Left: $\alpha'$ as a function of $\alpha$ averaged over all datasets. Right: The coverage ratio as a function of $\alpha$ averaged over all datasets.

Interestingly, the solutions produced by the *HMVE* and *I-OCSVM* methods for the largest approximated MV-set (associated with $\alpha_{19} = 0.95$) are equal to the solution of a single *OCSVM* algorithm trained with $\nu = 1 - \alpha_{19} = 0.05$. This equality derives from the definition of the *HMVE* and *I-OCSVM* methods. Therefore, in this setup, we claim that *q-OCSVM* also outperforms the *OCSVM* algorithm in the approximation of a single MV-set, and it does so with an average coverage ratio of 0.871 versus 0.821. We believe this improved performance is due to the parallelism constraint imposed by the *q-OCSVM* method on the hyperplanes, which acts as a regularization term on the solution. This observation is an interesting research direction to address in our future studies.

In terms of runtime complexity, our *q-OCSVM* method has higher computational complexity than *HMVE* and *I-OCSVM*, because we solve a global optimization problem rather than a series of smaller localized subproblems. However, with regard to the runtime complexity on test samples, our method is more efficient than *HMVE* and *I-OCSVM* by a factor of $q$, since the distances from each half-space only differ by their bias terms ($\rho_j$).

With regard to the choice of the Gaussian kernel width, parameter tuning for one-class classifiers, in particular for *OCSVMs*, is an ongoing research area. Unlike binary classification tasks, negative examples are not available to estimate the optimality of the solution. Consequently, we employed a common practice [1] of using a fixed width, divided by the number of features. However, in future studies, it would be interesting to consider alternative optimization criteria to allow tuning parameters with a cross-validation. For instance, using the average coverage ratio over all quantiles as an optimality criterion.

## 5.2   Experiments on Text Data

We evaluated our method on an additional setup of high-dimensional text data. We used the 20-Newsgroup document corpus [9]. 500 words with the highest frequency count were picked to generate

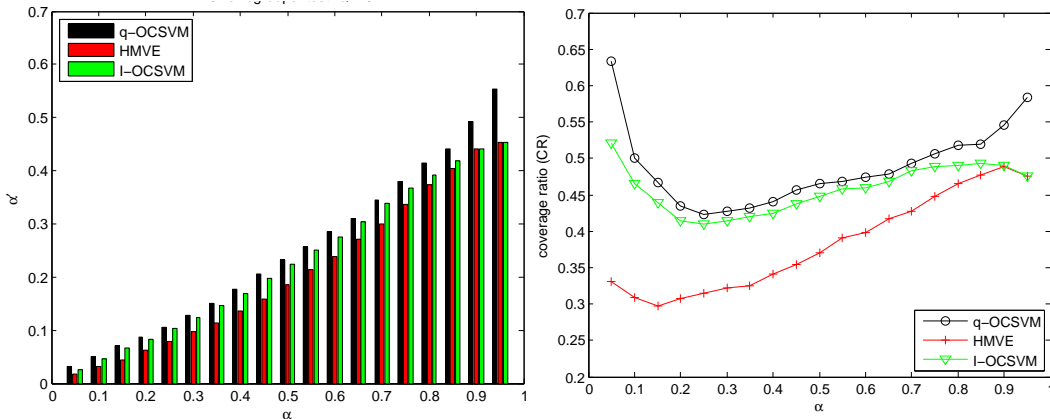

**Figure 3:** The *q-OCSVM*, *HMVE*, and *I-OCSVM* methods were trained to estimate 19 quantiles for the distribution of the 20 categories in the 20-Newsgroup document corpus. Left: $\alpha'$ as a function of $\alpha$ averaged over all 20 categories. Right: The coverage ratio as a function of $\alpha$ averaged over all 20 categories.

500 bag-of-words features. We use the sorted-by-date version of the corpus with 18846 documents associated with 20 news categories. From this series of documents, the first 100 documents from each category were used as the training set $\mathcal{X}$. The subsequent documents from the same category were used as the test set. We trained the reference methods with $\mathcal{X}$ to estimate 19-quantiles of a distribution, and evaluated the estimated $q$-quantiles with the test set.

Results on test data for each approximated MV-set are shown in Figure 3 in the same manner as in Figure 2 [10]. Unlike the experiments on the UCI repository, results in these experiments are not so close to the optimum, but still can provide useful information about the distributions. Again, our *q-OCSVM* method dominates the others with the best average $\alpha'$ and average coverage ratio behaviors. According to the Wilcoxon statistical test with $P < 0.01$, our method performs significantly better than the other competitors for each of the 19 quantiles separately.

It can be seen that the differences in coverage ratios between *q-OCSVM* and *I-OCSVM* in the largest quantile (associated with $\alpha_{19} = 0.95$) are relatively high, where the average coverage ratio for *q-OCSVM* is 0.555, and 0.452 for *I-OCSVM*. Recall that the solution of *I-OCSVM* in the largest quantile is equal to the solution of a single *OCSVM* algorithm trained with $\nu = 0.05$. These results are aligned with our conclusions from the UCI repository experiments, that the parallelism constraint, which acts as a regularizer, may lead to improved performance even for the approximation of a single MV-set.

## 6  Summary

The *q-OCSVM* method introduced in this paper can be regarded as a generalized *OCSVM*, as it finds multiple parallel separating hyperplanes in a reproducing kernel Hilbert space. Theoretical properties of our methods are analyzed, showing that it can be used to approximate a family of hierarchical MV-sets while preserving the guaranteed separation properties ($\nu$-property), in the same sense suggested by Schölkopf et al..

Our *q-OCSVM* method is empirically evaluated on a variety of high-dimensional data from the UCI repository and the 20-Newsgroup document corpus, and its advantage is verified in this setup. We believe that our method will benefit practitioners whose goal is to model distributions by $q$-quantiles in complex settings, where density estimation is hard to apply. An interesting direction for future research would be to evaluate our method on problems in specific domains that utilize $q$-quantiles for distribution representation. These domains include cluster analysis, outlier detection, and statistical tests.

## Footnotes

[1]In the following we call this method *I-OCSVM (independent one-class SVMs)*.

[2] A Gaussian kernel function $k(x_i,x_s) = e^{-\gamma||x_i-x_s||^2}$ was used in the following.

[3] Detailed setup parameters are discussed in Section 5.

[4]`archive.ics.uci.edu/ml/datasets.html`

[5]Nominal features were transformed into numeric ones using binary encoding; missing values were replaced by their features' average values.

[6]`http://web.eecs.umich.edu/˜cscott`

[7]The equivalent $C(\lambda)$ parameters of the *NOC-SVM* were initialized as suggested by the authors.

[8]In outlier detection, this measure reflects the ratio between expected and empirical false alarm rates.

[9]The 20-Newsgroup corpus is at `http://people.csail.mit.edu/jrennie/20Newsgroups`.

[10]Results for *NOC-SVM* were omitted from the graphs due to the limitation of the method in $q$-quantile estimation, which results in inferior performance also in this setup.

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
