[Reviews · NeurIPS 2013]

Submitted by Assigned_Reviewer_8

The authors present an interesting approach of estimating multiple one-class SVMs together instead of independently from each other. The paper is well written and structured.

- in (5)(6) the coupling between the different submodels is clear within the dual problem. However, it is less clear in the primal formulation. Please also explain how in (2) one can see the coupling between the submodels. On which of the unknown variables is this taking place?

- could the hierarchy constraint also be interpreted as a form of structural risk minimization?

- in Figure 1 a nice example is given with 4 hierarchical MV-sets. However, as a user here one should choose the number 4 beforehand. Is it also possible within the framework to treat this choice as a model selection problem? (as e.g. in clustering: Alzate C., Suykens J.A.K., Hierarchical Kernel Spectral Clustering, Neural Networks, vol. 35, Nov. 2012, pp. 21-30)

- Though the authors use many data sets, it is not clear how the kernel parameter has been selected (or any other parameters) in the different methods. This is a rather important aspect.

- It would be good to discuss differences also with the following references:

I. Steinwart, D. Hush, and C. Scovel, A classification framework for anomaly detection, J. Mach. Learn. Res., vol. 6, pp. 211-232, 2005.

A. Rakotomamonjy, M. Davy, One-class SVM regularization path and comparison with alpha seeding, Proc of 11th European Symposium on Artificial Neural Networks, Brugges, 2007

- There appears to be an error in the Gaussian kernel in footnote 2: ||.||^2 instead of (.)^2


The authors have carefully responded to the questions. I have updated my score accordingly.





Summary: Interesting approach of estimating multiple one-class SVMs in a coupled way. The model selection aspects should be clarified, also in making the comparisons.

Submitted by Assigned_Reviewer_9

Title: q-OCSVM: A q-Quantile Estimator for High-Dimensional
Distributions

Summary: This paper proposes an unsupervised method for learning the
quantile of a distribution. The idea is based on one class SVM. The
major contribution of the paper seems to be the fact that multiple
level of quantiles can be trained at once. The experiments on UCI
datasets suggest that q-OCSVM work better than training multiple
independent OCSVM.

The paper is well-written. Extending one-class SVM to deal with
multiple level of "novelty" seems to be a good idea. However, I feel
that the algorithmic contribution is too incremental. The experimental
results seem encouraging.

The bar graphs in Figure 3 and Figure 4 are difficult to read. Change
to color graphs maybe?

The choice of Gaussian width can be crucial. Fixing hyperparameters to certain
values is not convincing: it is important to perform some cross
validation over the hyperparameters.
Summary: An extension of one class SVMs for learning several quantiles. The experimental results seem encouraging but it could have been done more carefully.

Submitted by Assigned_Reviewer_10

The one-class SVM (OCSVM) algorithm, due to by Scholkopf et al, finds a hyperplane in feature space that maximizes the separation between the training set and the origin. The original OCSVM paper showed that, for suitable choices of kernel and slack variable penalty, the function returned by OCSVM is an indictor of the q-support* of the data distribution, i.e., a set of minimum volume that contains a q fraction of the distribution. A good way to characterize a distribution is to estimate its q-support for many values of q, and one could simply run OCSVM many times to generate these estimates -- this paper calls that algorithm I-OCSVM. However, these estimates won't necessarily be nested, while the actual q-supports must be. This paper describes an extension of the OCSVM algorithm, called q-OCSVM, for computing nested estimates of the q-supports of a distribution. Other papers have also proposed extensions of OCSVM with this property, but this paper's approach for enforcing the nesting constraint is very simple: the learned hyperplanes are required to be parallel, i.e., they only differ in their bias term. The paper modifies the OCSVM optimization problem to include this parallel constraint, and shows that the dual is still solvable in polynomial time for very high-dimensional feature spaces, via the kernel trick. The theoretical results in Section 4, which prove that the q-OCSVM algorithm asymptotically estimates the q-supports, are straightforward extensions of the results found in the original OCSVM paper. The experimental results show that the q-OCSVM algorithm outperforms existing algorithms, including I-OCSVM, for several datasets, though sometimes the improvement is small.

Some might regard the simplicity of the q-OCSVM extension as a reason not to accept the paper, but I view it as a point in the paper's favor. However, I do wonder whether an even simpler approach might work too. For example, I accept that forcing all the hyperplanes to be parallel acts as a kind of regularizer, and thus helps with overfitting. But couldn't this effect be achieved with the I-OCSVM algorithm by just increasing the penalty on the norm of the weight vector? Admittedly, this still would not guarantee that the estimated q-supports are nested. Which leads to a question: Does I-OCSVM intersect the q-supports that it learns, as suggested in the Introduction? Also, is q-OCSVM faster to train than I-OCSVM? It seems like it might be, since it's one QP and not several QPs. If so, that would be worth mentioning. Finally, instead of setting \nu = 1 - \alpha, what if we used a held-out validation set to choose a value for \nu that optimizes coverage ratio? It seems that might improve the results for all algorithms.

The results in Figures 2 and 3 are averaged over 61 and 21 datasets, respectively, but the plots do not contain error bars, so it's hard to judge the significance of the results. On the other hand, the fact that the q-OCSVM algorithm dominates the others for all values of q is encouraging. Also, I think the left- and right-hand side of each figure contain the same information, and thus seem redundant.

The paper is clear and well-written, and the relationship to previous work is well-explained. Figure 1 is a very helpful illustration. For completeness, it might be good to cite some work on estimating the quantiles of a _conditional_ distribution. Search for "nonparametric quantile regression" or "kernel quantile regression".

* The term "q-support" is mine, not the paper's.
Summary: The paper shows that a very simple extension of one-class SVM outperforms existing methods for computing nested estimates of all the q-supports of a high-dimensional distribution. I think it's an interesting contribution.
Author Feedback

Author rebuttal: With regard to the choice of the kernel width:
Parameter tuning for one-class classifiers, in particular for OCSVMs, is an ongoing research area. Unlike binary classification tasks, negative examples are not available to estimate the optimality of the solution. Consequently, we employed a common practice (Chih-Chung Chang and Chih-Jen Lin. LIBSVM: a library for support vector machines, 2011; Azmandian et-al, Local Kernel Density Ratio-Based Feature Selection for Outlier Detection, JMLR, 2012) of using a fixed width, divided by the number of features (mentioned in line 297). An approach similar to ours was also used in the original HMVE competing paper; a constant bandwidth was used by Lee and Scott (2011). In future studies, it would be interesting to consider alternative optimization criteria to allow tuning parameters with a cross-validation. For instance, as suggested by the reviewers, using the average coverage ratio over all quantiles as an optimality criterion. A note about this issue will be added to the paper.


With regard to the primal program:
Coupling between the submodels in the primal program (Equation 2) is done by summing $q$ OCSVM programs (Equation 1), while enforcing these programs to share the same $w$. As a result, the $q$ half-spaces in the solution of Equation 2 are different only by their bias terms, and thus parallel to each other. A clarification note will be added to the paper.


With regard to the questions on structural risk minimization and regularization:
Yes, we believe the hierarchical constraint acts as a regularizer in the $q$-OCSVM program, and thus the risk of overfitting in estimations can be reduced. Note that it is hard to obtain the same results by increasing the penalty on the norm in the I-OCSVM method, since penalties are not linked as in the hierarchical constraint, and thus their outcomes are less expected. Furthermore, it still won’t solve the absence of the $\nu$-property in the I-OCSVM solution.


With regard to the intersections of OCSVM solutions:
The problem of intersecting hyperplanes in the solution of OCSVMs is known in the literature: Lee and Scott (2011) discuss it and show an example that visualizes this problem; In the context of clustering problems, it was theoretically and empirically stated that OCSVM solutions are not hierarchically nested (Asa Ben-Hur, David Horn, Hava T Siegelmann, and Vladimir Vapnik. Support vector clustering. JMLR, 2002). As suggested by the reviewers, a note about this issue will be added to the paper.


With regard to the number of quantiles ($q$):
In our experiments we learned 19 quantiles since it nicely divides the [0,1] range to intervals of size 0.05 --- a common statistical unit of measure. We agree with the reviewer that it will be useful in some setups, such as in hierarchical clustering, to treat this choice of $q$ as a model selection problem. A note about this issue will be added to the paper.


With regard to error bars:
Note that our q-OCSVM was found significantly better for each quantile separately (Wilcoxon, 0.05), so that error bars provide no additional information. However, if the reviewers still find it useful, we will add it to the paper.


We thank the reviews for the additional references. We intend to include a discussion about them in the paper.